# Learning Good Policies by Learning Good Perceptual Models

## Abstract

Reinforcement learning (RL) has led to increasingly complex looking behavior in recent years. However, such complexity can be misleading and hides over-fitting. We find that visual representations may be a useful metric of complexity, and both correlates well objective optimization and causally effects reward optimization. We then propose curious representation learning (CRL) which allows us to use better visual representation learning algorithms to correspondingly increase visual representation in policy through an intrinsic objective on both simulated environments and transfer to real images. Finally, we show better visual representations induced by CRL allows us to obtain better performance on Atari without any reward than other curiosity objectives.

## 1 Introduction

In recent years, reinforcement learning(RL) has lead to increasingly complex behavior from simulated environments (Silver et al., 2016; OpenAI, 2018; Mnih et al., 2013; Andrychowicz et al., 2018). Yet despite this, there lacks a quantitative measure of intelligence in these agents. Qualitative measures can be deceptive. Consider agent Alice and Bob in Minecraft. Alice is capable of a constructing a house while Bob appears to only be able to navigate around the world. While at face value it may then appear that Alice is more complex, upon closer inspection we may find that Alice has simply memorized a set of actions to construct a house in that particular environment!

How can we be certain that our agents are not simply not memorizing a set of moves? One hypothesis is that the more intelligent an agent is, the more likely the inner representations in its policy will exhibit disentangled properties of the world. Towards this end, we investigate the emergent visual representations that occur in RL policies.

We investigate on various objectives and environment conditions, and find that the quality of visual representation learning correlates well with progress in reward optimization. Similarily, we find improved visual representations help agents perform better reward optimization. Thus, another natural question to ask is, how can we enable our agents to have better visual representations?

While there are ways to hardcode reward functions to enable agents perform well, can we come up with a generic objective that our agents can optimize that will directly lead them to have good representations? One idea towards this is to use recent work in curiosity. In curiosity, agents are typically given rewards corresponding to surprisal of state. But another view of curiosity is that of a minimax game where a curious agent is seeking to maximize the surprisal of an uncertainty model, while the uncertainty model seeks become less surprised about new states.

Thus, to enable a policy to learn good visual representations, we can treat the uncertainty model as a representation learning model. We then seek a policy that wants to lower the loss of the representation learning objective, while the model itself tries to optimize this loss. Under this objective, a policy must learn good visual representations, so that it is able to find visually surprising inputs for the vision model. We call this overall objective, Curious Representation Learning (CRL).

By coupling policy learning with representation learning, we find that CRL allows us to get better policy visual representations simply by applying better visual representation learning algorithms to the model. As a result, we find that CRL obtains consistently good representations in policies across environment size and type, often beating many hard-coded domain specific objectives. As an added bonus, we find that CRL is also able to achieve better visual representation learning than other data collection methods, as it actively sees diverse inputs that surprise it.

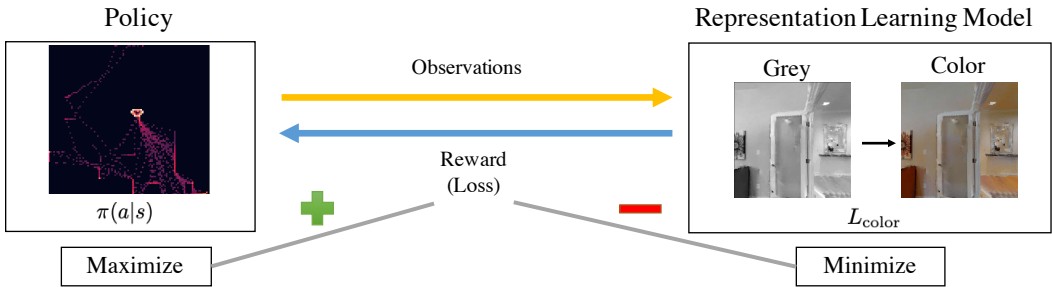

Figure 1: Overview of CRL (curious representation learning). We use the loss of a representation learning algorithm as an intrinsic reward for an agent. Correspondingly, we train the representation learning algorithm on the experience generated by the agent. This forces the agent and model compete in minimax game to learn good visual representations

In addition, we investigate optimizing visual representation through CRL as a curiosity bonus. We find that on Atari, CRL is able to obtain better overall performance compared to other approaches such as Forward Dynamics and RND, when used as a sole intrinsic bonus.

Our contributions are threefold. 1) We first show that visual representation learning corresponds well to reward optimization in RL policies. 2) We propose CRL, a method that allows us to use better visual representation algorithms to obtain better visual representation in policies on simulated environments and real images. 3) We apply CRL to Atari and show that the better visual representation obtained in policies allows us to out-compete forward dynamics (Burda et al., 2018a) and Random Network Distillation benchmarks (Burda et al., 2018b).

## 2 RELATED WORK

Visual representation learning has seen a large amount of interest in recent years (Zhang et al., 2016; Gidaris et al., 2018; Hjelm et al., 2018; Bengio et al., 2013). Approaches towards unsupervised learning include predicting image rotations (Gidaris et al., 2018), colorizing images (Zhang et al., 2016), maximizing mutual information between observations and representations (Hjelm et al., 2018), and many other approaches. Much of the previous visual representation work has focused on a passive datasets – here we focus on representation learning in active environments where agents must interact with the environment to obtain data.

Curiosity has also been studied extensively in the past years (Houthooft et al., 2016; Pathak et al., 2017; Bellemare et al., 2016; Burda et al., 2018b). Such objectives typically encourage agents to minimize some uncertainty in the world, based off dynamics (Houthooft et al., 2016; Pathak et al., 2017) or random features (Burda et al., 2018b). Our proposed method for curiosity differs from past method as we propose curiosity as a minimax game between a generic representation learning algorithm and a target policy. This formulation can help explain some of the strong results of curiosity as well as pave a direction towards better curiosity algorithms.

Our work also relates to past work in active learning. Active learning involves sampling datapoints that maximize an uncertainty/error measure, while the learner itself thereby aims to minimize uncertainty/error (Settles, 2009; Roy and McCallum). This entails an adversarial game where as learners get better and better, it becomes increasingly difficult to train on, an idea refered to as curriculum learning (Bengio et al., 2009). In our work, explicitly formulate this arms race as an explicit minimax optimization problem with a representation learning model.

## 3 FORMULATION

In this section we describe our approach towards achieving/measuring visual representations. We describe visual representation learning and associated algorithms in Section 3.1 and discuss curious representation learning (CRL) in Section 3.2. We further detail our architectures in Section 3.3 and describe our environment setup in Section 3.4.

### 3.1 VISUAL REPRESENTATION LEARNING

To evaluate visual representation learning, we test disentanglement of features at higher layers in neural network. To test this, we train a linear map on a frozen feature embedding from a backbone architecture specified in Section 3.3 on a classification task, and report test set accuracy as the visual representation value. We note that Kolesnikov et al. (2019) shows that linear maps are an adequate

overall disentanglement of features representation learning. We consider 3 different approaches towards visual representation learning which we detail below.

**Colorization** Zhang et al. (2016) propose to predict a color of image from its input luminance. Intuitively, by learning accurate to color images, a model must be able to have good visual representation of objects.

**Autoencoding** Autoencoding tries to reconstruct an image through a bottleneck Bengio et al. (2013) . By being forced to compress an image into a compressed representation, a model should learn good visual representation of objects.

**RND** Burda et al. (2018b) propose to train a network to predict random features obtained from another network. Such a task requires a model to learn behavior of another model, and we surprisingly find this leads reasonable representation learning on VizDoom environments.

We also evaluated rotation prediction based off (Gidaris et al., 2018), but found that it performed poorly in ViZDoom, due to obvious rotation artifacts.

On the ViZDoom environment, our classification task is a curated dataset of different VizDoom objects (monsters, health packs, and guns). For the Habitat environment, our classification task is room scenes from the Places365 dataset (Zhou et al., 2017).

### 3.2 CURIOUS REPRESENTATION LEARNING

When training agents in reinforcement learning, an agent receives a reward $r_t$ at timestep $t$ and seeks to produce a policy $\pi(s_t; \theta_p)$, represented by a neural network that maximizes the expected overall reward

$$\max_{\theta_p} \mathbb{E}_{\pi(s_t;\theta_p)}[\Sigma_t r_t]. \tag{1}$$

With curiosity, this reward can be decomposed into components $r_i^e$ which is extrinsic and $r_i^i$, which is an intrinsic reward that is dependent on separate model $\theta_m$, so that we seek parameters $\theta_m, \theta_p$, so that we have

$$\max_{\theta_p} \mathbb{E}_{\pi(s_t;\theta_p)}[\Sigma_t f_{\theta_m}(s_t)]; \quad \min_{\theta_m} \mathbb{E}_{\pi(s_t;\theta_p)}[\Sigma_t g_{\theta_m}(s_t)] \tag{2}$$

for some surprisal objective $f_{\theta_m}$ and modeling objective $g_{\theta_m}$. This suggests that by setting a surprisal objective $f_{\theta_m}$ to be the same as the modeling objective $g_{\theta_m}$, we obtain a Minimax objective

$$\max_{\theta_p} \min_{\theta_m} \mathbb{E}_{\pi(s_t;\theta_p)}[\Sigma_t g_{\theta_m}(s_t)] \tag{3}$$

between an agents policy $\theta_p$ and some world model $\theta_m$.

The above formulation of curiosity now gives us a natural method to improve visual representation learning in both policy and models. By replacing $g_{\theta_m}(s_t)$ with the loss of a generic visual representation learning algorithm $L_{\text{rep}}$, we can now simultaneously make both the models and policies have good visual representation in an environment. This is because the representation learning model $\theta_m$ is able learn on more diverse data from a policy optimized explore data that confuse the model, while the policy $\theta_p$ is forced to focused to learn good visual representations to confuse the chosen representation learning model. We provide an illustration of the above object in Figure 1.

### 3.3 ARCHITECTURE SETUP

To balance capacity and enable comparison between representation learning models and between policies, we use identical base architecture for both RL policies and representation learning models. For both, we use 7x7 convolution with stride, followed by a max-pooling layer, followed by 2 residual blocks (with down-pooling), with residual blocks having 64 and 128 filters. We then evaluate linear classification performance performance after flattening the 128 filter residual block. Models are trained and evaluate on 168x168 images.

For reinforcement learning policies, when training the base architecture, we flatten observations from the 128 filter residual block and then apply an LSTM cell to which we apply MLPs to decode action and value heads. We use proximal policy optimization (Schulman et al., 2017) to train our policies, using the default hyper-parameters from Atari Environments. In representation learning models, for RND, we flatten then residual block and then apply an MLP to predict the random target. For colorization, we apply 3 additional convolutions to the 128 filter residual block to colorize. For autoencoding, we use a 128 hidden units bottleneck with 2 residual blocks of upsampling.

3.4 ENVIRONMENTS

We evaluate visual representations on two different visual environments:

**VizDoom**   We first evaluate on the ViZDoom (Kempka et al., 2016) environment. We use the a random level generator Oblige to generate diverse different Doom environments and report configuration details in the appendix. Agents have the ability to move forward, backwards, left and right, turn left and right, attack, speed, and interact with different objects.

Agents are spawned in maps with a large number of rooms (greater than 10), with a starting room and ending room through which agents must interact to leave/enter. There are a large number of monsters scattered across the rooms, and agents must either avoid or shoot monsters, as well avoid hazardous traps.

We evaluate agent performance on varying numbers of randomly generated training maps. Environments with different number of maps assess our visual representation learning algorithms in different ways. On a single environment, to obtain the best visual representation, it is necessary to explore over the entire space. In contrast, in the setting with a large number of environments, to obtain a good visual representation, agents may need just a directed way to explore the environments. We compare with a set of designed objective listed below:

1. (Counts Based): Agents are given a counts based reward based off 1 divided by the square root of the visitation count at the state. States are based off a discretized grid and discretized rotation requirement absolute position and orientation of an agent.

2. (Movement Distance): Agents are given a reward based off the maximum distance from start location. Agents are given a positive reward every time it moves further from the origin than before and otherwise returns 0 reward.

3. (Shooter Optimization): Agents are given a reward based off damage dealt to enemies, number of enemies killed, and any health gained.

**Habitat**   We further investigate visual representation learning in the Habitat (Manolis Savva* and Batra, 2019) environment, using the Matterport and Gibson house scans. The Habitat simulator consists of sets of rooms in houses. Under this setting, agents have the ability to move forward or turn left and right. We compare performance with the explicit objective called PointNav (Manolis Savva* and Batra, 2019), in which agents are given a relative distance to a goal they must reach.

**Atari**   As a benchmark for measuring the intrinsic curiosity, we also evaluate on the Atari benchmark, using CRL as the only intrinsic reward. We evaluate on Atari environments of BeamRider, Breakout, Montezuma's Revenge, Pong, Abert, Riverraid, Seaquest and SpaceInvaders as done in (Burda et al., 2018a).

## 4 EXPERIMENTS

We provide empirical and qualitative evaluations of our hypothesis. First, we evaluate the correspondence between policy learning and the visual representation learning of the policy in Section 4.1 on different objectives. Next we evaluate visual representations that emerge in CRL on both synthetic environments and real images. Finally, we evaluate CRL's ability to enable good performance on Atari with no extrinsic reward in Section 4.3.

### 4.1 HOW DOES VISUAL REPRESENTATION LEARNING CORRESPOND WITH POLICY PERFORMANCE?

We measure the relationship of visual representations and policy reward on ViZDoom (Section 3.4). We systematically varied the number of different training levels generated to train policies with different complexities. We see from Table 1 that across different objectives and different environment numbers (as a proxy of environment complexity), there is a strong correlation that is consistent between both the criterion and environment number.

We provide a example visualization of the trend of representation learning with score optimization in Figure 2. In general, we find that visual representations correlate well with initial increases with objective (steps before 1e7). However, once the maximum score is achieved and training has plateaued, then the visual representation correspondingly decreases slightly (steps after 1e7), suggesting that it may be beneficial to stop RL policy training once reward performance stops.

We further investigated the opposite effect on whether better visual representations would lead to better objective optimization. To test this, we optimize the movement distance training objective

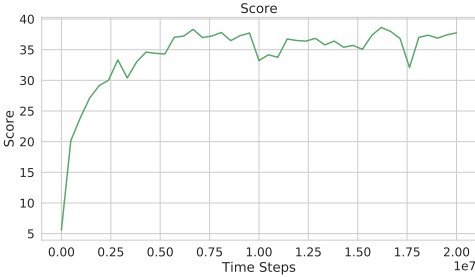 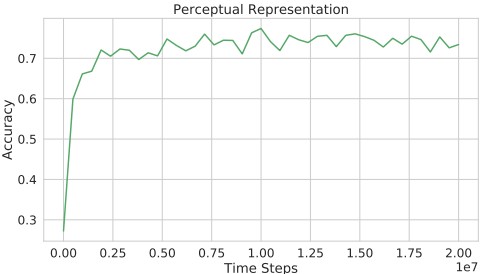

Figure 2: An visualization of objective optimization with corresponding perceptual representation learning on the maximum distance from origin objective with 10 environments.

| Environment Number | Counts Based | Movement Distance | Shooter Opt. |
|---|---|---|---|
| 1 | 0.794 | 0.719 | 0.573 |
| 10 | 0.911 | 0.733 | 0.549 |
| 100 | 0.970 | 0.922 | 0.647 |
| 1000 | 0.689 | 0.826 | 0.635 |

Table 1: Correlation coefficient of measured representation accuracy and objectives (reward across different environment numbers in ViZDoom evaluate across 3 different seeds). P values are all in the 1e-6 - 1e-12 range.

| CRL Model | Training Reward | Classification Performance |
|---|---|---|
| Autoencoding | 47.63 | 0.799 |
| RND | 15.61 | 0.721 |
| Colorization | 29.69 | 0.766 |
| Random | 19.61 | 0.468 |

Table 2: Relative performance on the Movement Distance (10 levels) objective when using CRL initialized with curious policies with different representation learning values indicated compared with random policy after 1M frames of training.

with policies initialized with different visual representations from CRL. We report the the relative magnitude of overall reward compared with initializing from random policy after 1 million frames of training in Table 2. Overall, we find that better visual representations obtained from CRL are able to lead to better relative performance after 1 million frames, with policy initialization with CRL with an autoencoding objective outperforming a policy from scratch by 242% in relative performance.

This result suggests that better visual representations both naturally emerge through RL policy training, and further may be a generic objective to optimize to lead to faster reinforcement learning on many different tasks.

### 4.1.1 WHAT VISUAL REPRESENTATIONS EMERGE FROM CURIOUS REPRESENTATION LEARNING?

| Environment Number | Policy | | | Representation Model | | |
|---|---|---|---|---|---|---|
| | Autoencode | Colorization | RND | Autoencode | Colorization | RND |
| 1 | **0.748** (0.016) | 0.676 (0.022) | 0.672 (0.014) | **0.864** (0.014) | 0.746 (0.018) | 0.751 (0.024) |
| 10 | **0.794** (0.005) | 0.733 (0.010) | 0.746 (0.013) | **0.872** (0.002) | 0.742 (0.017) | 0.789 (0.020) |
| 100 | **0.811** (0.006) | 0.762 (0.007) | 0.810 (0.012) | **0.863** (0.013) | 0.767 (0.010) | 0.792 (0.020) |
| 1000 | **0.819** (0.011) | 0.747 (0.005) | 0.806 (0.011) | **0.855** (0.002) | 0.770 (0.002) | 0.793 (0.019) |

Table 3: Correspondence between both better representations in policies and representation learning models (RND = random network distillation) under CRL. Value evaluated across 3 different seeds with standard error in parentheses. These result show stronger visual representation learning algorithms give stronger visual representations in policies.

We next analyze the effect of CRL in inducing good visual representations. In Figure 4 we show a scatter plot of policy representation and model representation under CRL. In Table 3, we provide quantitative numbers of visual representations from CRL on both policies and representation learning models. We find that better representation learning algorithms lead to significantly better policy representations, with autoencoding being the best representation learning algorithm in VizDoom. We see identical rankings between policy visual representations and representation learning models visual representations. These results help suggest that the search for generic representation learning models can corresponding lead to policies with even better visual representations.

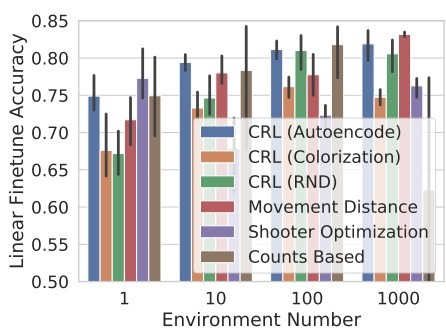
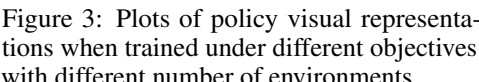
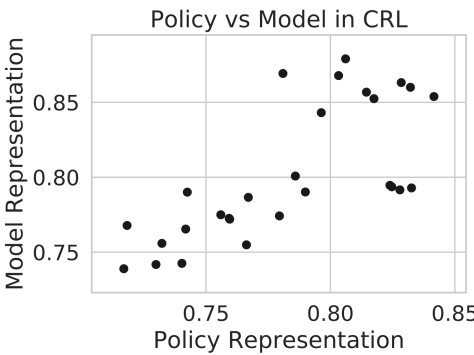

Figure 3: Plots of policy visual representations when trained under different objectives with different number of environments

Figure 4: Plots of policy visual representations with representation model visual representation under CRL from 10, 100, 1000 environments. There is a good correlation between policy representation and model representation

| Environment Number | Counts Based | Movement Distance | Shooter Optimization | CRL Autoencode | Colorization | RND |
|---|---|---|---|---|---|---|
| 1 | **0.749** (0.028) | 0.717 (0.017) | 0.772 (0.019) | 0.748 (0.016) | 0.676 (0.022) | 0.672 (0.013) |
| 10 | 0.783 (0.046) | 0.780 (0.010) | 0.679 (0.025) | **0.794** (0.011) | 0.733 (0.010) | 0.746 (0.010) |
| 100 | **0.818** (0.019) | 0.778 (0.013) | 0.723 (0.010) | 0.811 (0.012) | 0.762 (0.007) | 0.810 (0.012) |
| 1000 | 0.623 (0.079) | **0.832** (0.002) | 0.763 (0.006) | 0.819 (0.021) | 0.747 (0.005) | 0.805 (0.012) |

Table 4: Comparison of visual representations learned in policies from CRL and other objectives. We find that CRL with an autoencoding objective consistently gives relatively good visual representations while other objective, such counts based may sporadically give better visual representation, but are not consistently across environments.

| Environment Number | Counts Based | Movement Distance | Shooter Optimization | CRL | Random Policy |
|---|---|---|---|---|---|
| 1 | 0.858 (0.008) | 0.862 (0.003) | 0.820 (0.005) | **0.864** (0.014) | 0.834 (0.012) |
| 10 | 0.865 (0.006) | 0.862 (0.001) | 0.851 (0.004) | **0.872** (0.003) | 0.855 0.002 |
| 100 | 0.862 (0.002) | 0.860 (0.001) | 0.847 (0.002) | **0.863** (0.002) | 0.853 (0.003) |

Table 5: Table of visual representations learned from an autoencoding model from data collected from different policies trained in different number of ViZDoom environment. CRL incentives the policy to generate diverse data that allows the best model representations across different environment numbers.

In Table 4, we compare visual representations learned in policies trained on fixed objectives to those that emerge in CRL. We provide a qualitative graph in Figure 3. We find that CRL with an autoencoding objective obtains the best or close to the best visual representations in policies across different environment. While on different environments numbers, either shooter optimization, movement distance, or count based may obtain slightly better policy visual representations, CRL with autoencoding objective has significantly lower variance in observed visual representations, and performs consistently well across different environment numbers. Furthermore, we note that CRL does not require any specification of source of task and works generically on any environment, while other objective may require significant manual specification.

We also compare visual representations of representation learning models under different objectives in Table 5. For non CRL objectives, we train an autoencoding model (best performing representation learning model on VizDoom) on data collected by a policy. We compare CRL model representations with autencoding models trained on data from policies initialized from scratch or trained on counts based, movement distance, shooter optimization, and CRL objectives. We find, regardless of the environment number, that CRL allows the representation learning model to have best possible visual representation in the environment.

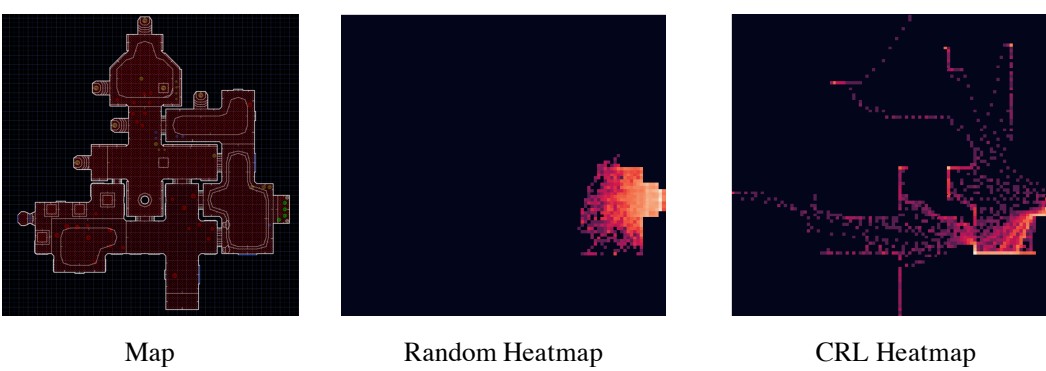

| Map | Random Heatmap | CRL Heatmap |

Figure 5: Comparison of random policy with CRL (curious representation learning) on a test level. CRL (with autoencoding objective) is able to induce diverse paths that explore different sets of rooms.

We further qualitatively evaluate the effectiveness of CRL in inducing exploration of the surrounding environment. Figure 5 shows that CRL is able to lead to effective exploration schemes throughout a large mapped environment compared to random policy and a test level.

### 4.2 Do Visual Policy Representations Transfer to Real Images?

We next evaluate on Habitat (Section 3.4) to see if the visual representations learned through CRL transfer to real images from rooms in the Places dataset. We compare representations learned through CRL on RND, autoencoding, and colorization objectives as well as the PointNav objective.

Random CRL

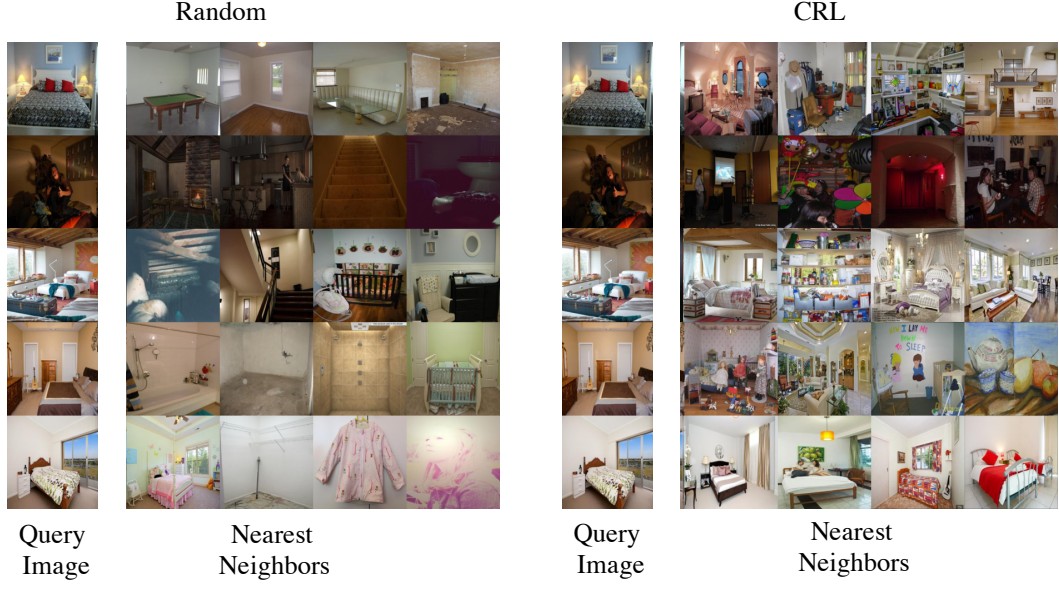

| Query Image | Nearest Neighbors | Query Image | Nearest Neighbors |

Figure 6: Illustration of nearest neighbors on room scene in Places of a CRL model trained on Habitat compared to a random network. CRL training in simulation transfers to real scene as seen in the beds in the last row.

On real world images, similar to the VizDoom enviroment, we find that CRL is able to enable good visual features in Table 6. Trends between better model and visual representation hold, similar to in VizDoom with CRL, with the most effective representation learning algorithm being colorization. We find that CRL enables us to train representation learning models and policies to have **significantly better** visual features then agents optimizing the PointNav goal, with linear classification accuracy of 0.193 compared to 0.086 in PointNav for policies and 0.253 compared to 0.211 for the representation learning model. These linear classification accuracies are significantly better than random (0.084) and somewhat close to the linear classification accuracy of a colorization model directly trained on Places room scenes (0.324).

| - | PointNav | CRL (AE) | CRL (CL) | CRL (RND) | CL (oracle) | Random |
|---|---|---|---|---|---|---|
| Policy | 0.086 | 0.175 | 0.193 | 0.164 | - | - |
| Model | 0.211 | 0.1066 | 0.253 | 0.085 | 0.324 | 0.084 |

Table 6: Comparison on linear finetuning classification accuracy on room scenes in Places using policies and representation learning models (using colorization or specified CRL objective) trained from data from Point Navigation and different CRL objectives. We also compare with a colorization model trained on data from Places room scenes and a randomly initialized model.

To qualitatively study the visual representations in Habitat, we construct nearest neighbors in embedding space of a trained CRL colorization model and a random model in Figure 6. We find that CRL trained models on Habitat are able cluster certain images in Places room scenes together such as beds.

### 4.3 DOES INCENTIVIZING BETTER VISUAL REPRESENTATIONS LEAD TO MORE CURIOUS BEHAVIOR?

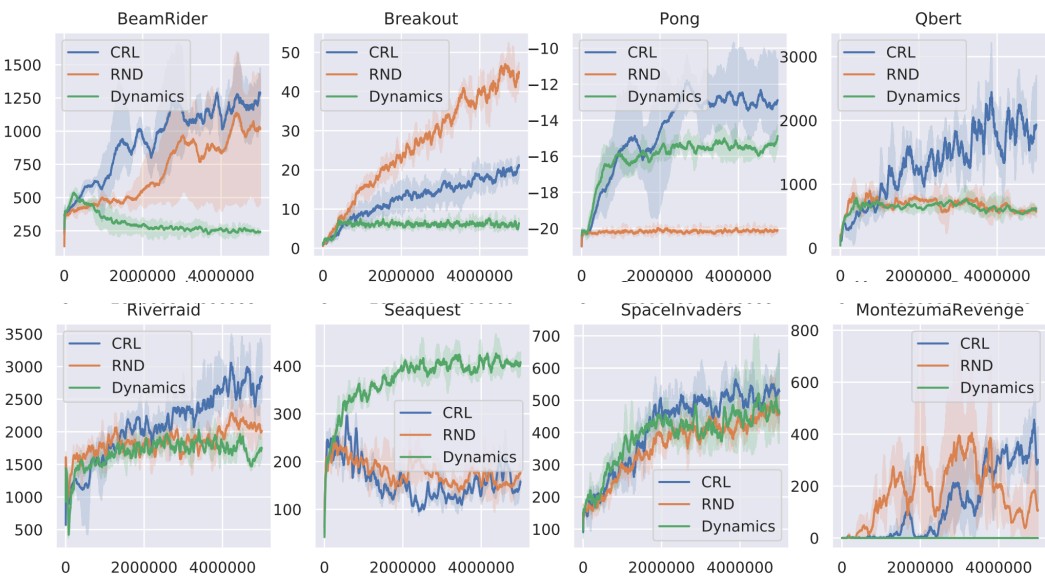

Figure 7: Comparison of CRL (curious representation learning with autoencoding objective) vs RND vs Dynamics on Atari using only intrinsic reward across 3 different seeds. CRL performs favorably and gets the highest score in 6 of the 8 evaluated environments.

Next we investigate whether using CRL as a intrinsic reward gives better performance than other curiosity models, We compare with Random Network Distillation and Forward Dynamics in Figure 7 across 3 different random seeds. We choose the autoencoding objective in CRL for the representation learning objective, as we find it gets the best visual representations on VizDoom.

Overall, we find that CRL performs favorably compared to both RND and forward dynamics, and gets the best overall score on 6 of the 8 evaluated environments. This suggests that obtaining best visual representations may also be a manner to improve curiosity.

## 5 CONCLUSION

In this paper, we have shown visual representations correspond and help reward optimization. Motivated by this insight, we propose a new method, CRL, that allows us to get improved visual representations in policies through better visual representations in model. We further illustrate that these better visual representation can provide incentives to explore more in no reward scenarios. We hope that our results will inspire further exploration on both better visual representation learning models/policies and better reward optimization.

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

# A APPENDIX

## A.1 DOOM SETUP

We use the following setup for setting the Oblige random map generator.

| Configuration | Value |
|---|---|
| Length | Regular |
| Size | Regular |
| Health | Normal |
| Weapons | Very Soon |
| Theme | Jumble |
| Game | Doom 2 |
| Mons | More |
| Ammo | More |
| Strength | Harder |
| Outdoors | Mixed |
| Doom Level | 5 |

Table 7: List of configurations used for the Oblige game engine.

## A.2 NEAREST NEIGHBORS DOOM

Random                                                     CRL

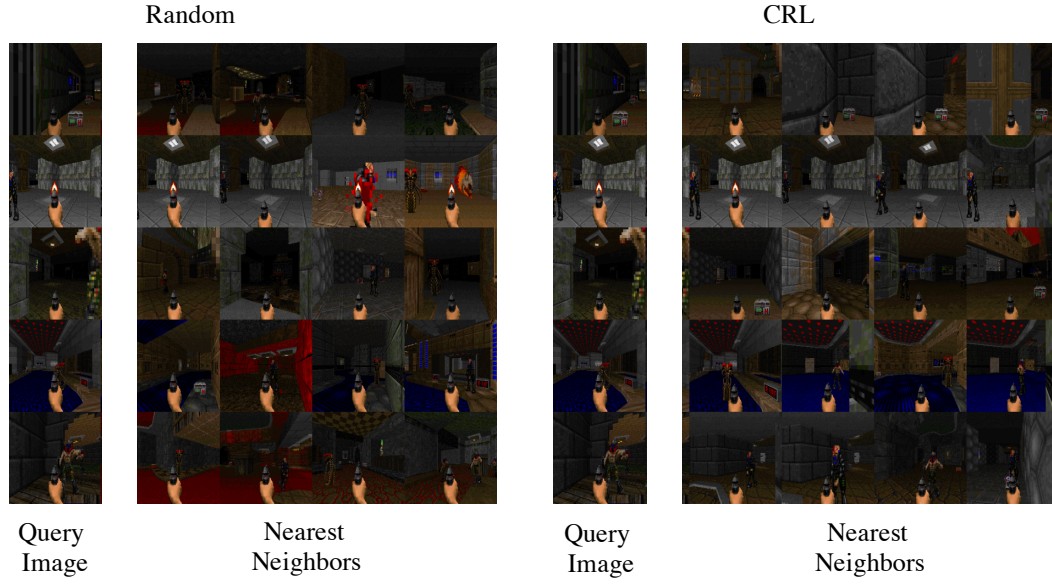

Query            Nearest            Query            Nearest
Image           Neighbors           Image           Neighbors

Figure 8: Illustration of nearest neighbors in Doom of a CRL(AE) policy trained on Habitat compared to a random network. Nearest neighbor in CRL space is able to cluster more visually similar images

We further show nearest neighbor images on VizDoom in Figure 8. The leftmost column is the query image while the other 4 columns are the 4 nearest neighbors in embedding space. Training through CRL allows clustering of various doom objects.

