# OpenReview forum: "Learning Good Policies By Learning Good Perceptual Models"
_ICLR.cc/2020/Conference — Reject_

### Official Review · AnonReviewer2 · 2019-10-22
**Official Blind Review #2**

**Rating:** 1

**Review:**

This paper presents an empirical study of using error reduction as a curiosity measure. The authors consider an auto-encoder model, a colorization model and RND as intrinsic motivation signals. I find the write up very unclear and have trouble understanding what the claims are and how they are backed up.

Major points:
* About the claims as stated on page 2: 1) The first claim I don't understand, I think what is meant is on navigation tasks they find "their measure of representation learning" (proposed in kolesnikov 2019) seems to correlate well with reward optimization. In 3.1 to back this claim they claim to test disentanglement but seem to test classification. I see no reason to not put that part in 4.1. 2) The authors claim to propose a new method: it can't be a separate representation network to derive rewards because that is what burda et al. and many many others do, it can't be the minimax formulation because that has been known for a while (e.g. predictability minimization schmidhuber) so I am not sure what the claim is about. What exactly is novel about the model. 3) CRL seems at best to outperform baselines on beamrider, qbert and riverraid but the results are impossible to assess. We don't know what the x axis is in figure 7 (is it frames, with or without repeats, is updates etc). Pong -12 is far from learnt for instance and its one of the easiest games.
* suprisal objective and modeling objectives are very high level concepts that we can talk about in the introduction and conclusion but much more precise terms need to be employed in the model exposition. The readers need to know what sort of properties they should have ideally easily identify examples (without having to read 2 papers and a large survey). I would start with the minimax formula and then explain what is considered, how the different intrinsic rewards are added, if they are normalized, how they are weighted etc. etc.
* the details about failed experiments, environments and architecture should IMO be relegated to experiments
* It should be very clear early on that the model is separate from the representation.
* auto-encoders are a large family of models and it is not clear from the paper which exact model is meant by the authors. Also, citing Bengio 2013 is NOT a valid citation for auto-encoders. The right citation depends on the model you apply please use that!


**Experience Assessment:**

I have published one or two papers in this area.

**Review Assessment: Checking Correctness Of Derivations And Theory:**

I assessed the sensibility of the derivations and theory.

**Review Assessment: Checking Correctness Of Experiments:**

I assessed the sensibility of the experiments.

**Review Assessment: Thoroughness In Paper Reading:**

I read the paper at least twice and used my best judgement in assessing the paper.

---

### Official Review · AnonReviewer1 · 2019-10-23
**Official Blind Review #1**

**Rating:** 3

**Review:**

This paper formulates curiosity based RL training as learning a visual representation model, where the policy tries to maximise the loss of a shared visual model minimising an auxiliary task (such as autoencoding the input).

Curiosity is an important topic in the RL field and this paper is well motivated. I also like the approach taken as it looks into this problem through the lens of better representation learning (LR) and arguing that with focusing on better LR and maximising model loss for novel scenes, we are going to get also better overall performance.

However, there are a few key question marks that are still open and I would suggest them to be answered explicitly in the paper:

1) What is the relationship with methods that use auxiliary tasks for unsupervised training in RL (e.g. Jaderberg et al, ICLR 2017)? It's clear that this method doesn't use any extrinsic reward function but the underlying architecture is similar.

2) Similar to above, the comparisons and contrasts to Burda et al, ICLR 2019 could be made more explicit as the objective functions such as autoencoding which seems to be working well in this paper has also been studied in that work.

3) Continuing with comparisons, it's not clear if this method delivers better performance compared to other curiosity based methods. For examples, the top scores in Fig 7 are considerably lower than those achieved in Burda et al, ICLR 2019 (Fig 2). Similarly, we don't know how the method compares to state-of-the-art on other tasks considered in the paper. As a result, the paper lacks good benchmarking against state-of-the-art in this space and discussion on pros and cons.

4) It seems that in Tab 1 the correlation collapses for the last row, any reason why this is happening?

5) It would be good to add both a system diagram as well as a network architecture to clarify how everything is wired.

6) The training details are missing, both in terms of hyperparameters as well as optimisation strategy for solving minmax.

7) Minor: RND is used in the experimental section to refer to both random feature prediction and random network distillation, so would be better to use different references.

**Experience Assessment:**

I have read many papers in this area.

**Review Assessment: Checking Correctness Of Derivations And Theory:**

I assessed the sensibility of the derivations and theory.

**Review Assessment: Checking Correctness Of Experiments:**

I carefully checked the experiments.

**Review Assessment: Thoroughness In Paper Reading:**

I read the paper thoroughly.

---

### Official Review · AnonReviewer3 · 2019-10-24
**Official Blind Review #3**

**Rating:** 1

**Review:**

This paper proposes a framework called curious representation learning (CRL) which uses a better visual representation in RL. They show that better visual representation helps reward maximization.

I have to recommend rejection for this paper. It appears 1) the idea of using curiosity is not originated from this paper; 2) I do not see what is a "better visual representation"; 3) the comparison with baselines does not show that the new method is consistently better.

The paper is also very hard to read. I would think the name "curious representation learning" means "representation learning is curious". There are many inaccurate languages used in the paper. To list a few: "complex behavior", "in curiosity", "disentanglement"... I do not really understand what does it mean.

**Experience Assessment:**

I have published in this field for several years.

**Review Assessment: Checking Correctness Of Derivations And Theory:**

I assessed the sensibility of the derivations and theory.

**Review Assessment: Checking Correctness Of Experiments:**

I assessed the sensibility of the experiments.

**Review Assessment: Thoroughness In Paper Reading:**

I read the paper at least twice and used my best judgement in assessing the paper.

---

### Decision · Program_Chairs · 2019-12-19

**Decision:**

Reject

**Comment:**

This paper investigates using "curiosity" to improve representation learning. This paper is not ready for publication. The main issues was the reviewers found the paper did not support the claim contributions in terms of (1) evaluating the new representations and improvement due to the representation, and (2) the novelty of the method compared to the long literature in this area. In general the reviewers found the empirical evidence unconvincing, and the too many missing details.

The results in this paper have many issues: claims of performance based on three runs; undefined error measures; bolding entries in tables which appear not significantly better without explanation; unclear/informal meta-parameter tuning.

Finally, there are some terminology issues in this paper. I suggest an excellent paper on the topic: https://www.ncbi.nlm.nih.gov/pmc/articles/PMC3858647/